# Rapid whole-brain electric field mapping in transcranial magnetic stimulation using deep learning

Guoping Xu[1,2], Yogesh Rathi[2,3,4], Joan A. Camprodon[3,4], Hanqiang Cao[5], Lipeng Ning [2,3,4]*

**1** School of Computer Sciences and Engineering, Wuhan Institute of Technology, Wuhan, Hubei, China,
**2** Department of Psychiatry, Brigham and Women's Hospital, Boston, MA, United States of America,
**3** Department of Psychiatry, Massachusetts General Hospital, Boston, MA, United States of America,
**4** Harvard Medical School, Boston, MA, United States of America, **5** School of Electronic Information and Communications, Huazhong University of Science and technology, Wuhan, Hubei, China

* lning@bwh.harvard.edu

**Data Availability Statement:** All data used in this study are freely available from public website. Results and code from this paper are available at https://github.com/LipengNing/Efield.

**Funding:** This work was supported in part by NIH grants R21MH126396 (LN), R21MH115280 (LN

## Abstract

Transcranial magnetic stimulation (TMS) is a non-invasive neurostimulation technique that is increasingly used in the treatment of neuropsychiatric disorders and neuroscience research. Due to the complex structure of the brain and the electrical conductivity variation across subjects, identification of subject-specific brain regions for TMS is important to improve the treatment efficacy and understand the mechanism of treatment response. Numerical computations have been used to estimate the stimulated electric field (E-field) by TMS in brain tissue. But the relative long computation time limits the application of this approach. In this paper, we propose a deep-neural-network based approach to expedite the estimation of whole-brain E-field by using a neural network architecture, named 3D-MSResUnet and multimodal imaging data. The 3D-MSResUnet network integrates the 3D U-net architecture, residual modules and a mechanism to combine multi-scale feature maps. It is trained using a large dataset with finite element method (FEM) based E-field and diffusion magnetic resonance imaging (MRI) based anisotropic volume conductivity or anatomical images. The performance of 3D-MSResUnet is evaluated using several evaluation metrics and different combinations of imaging modalities and coils. The experimental results show that the output E-field of 3D-MSResUnet provides reliable estimation of the E-field estimated by the state-of-the-art FEM method with significant reduction in prediction time to about 0.24 second. Thus, this study demonstrates that neural networks are potentially useful tools to accelerate the prediction of E-field for TMS targeting.

## Introduction

Transcranial magnetic stimulation (TMS) is a non-invasive neuromodulation technique increasingly used to study human physiology, cognition, brain-behavior relations and the pathophysiology of neurologic and psychiatric disorders [1]. In TMS, a magnetic coil with

and JC), K01MH117346 (LN), R21MH116352 (LN), R01MH112737(JC). XG was supported by Graduate School of Huazhong University of Science and Technology. This study was partially completed while XG visited the Psychiatry Neuroimaging Laboratory at Brigham and Women's Hospital in 2019. No additional external funding was received for this study.

**Competing interests:** The authors have declared that no competing interests exist.

pulsed current is placed over the scalp to generate an electric field (E-field) in the underlying brain tissue in order to modulate neural activity in a target brain region [2]. However, due to the complex structure of the brain and the electrical conductivity variation across different tissues, the maximal stimulation strength of E-field does not always happen at the expected location [3] and the current pathways induced by electrical stimulation are not straightforward to identify [4]. Thus, it is necessary to accurately estimate the distribution of electric field (E-field) induced in the brain to improve brain targeting in TMS [5].

Several numerical approaches have been developed to estimate the E-field in TMS, including the local sphere model [6, 7], the boundary element method (BEM) [8, 9] and the FEM method [10–12]. In particular, the FEM method is based on a volume conductor model (VCM) of head tissue which not only is able to characterize complex tissue structure, brain geometry but also anisotropic tissue conductivity to potentially improve the model precision especially in white matter regions [13, 14]. Moreover, the FEM approach is able to integrate individual anatomical head models (volume conductor models) using several standard toolboxes such as SimNIBS [15], FreeSurfer [16], FSL [17], and SPM [18], to estimate the spatial distribution of stimulated tissue taking into account the impact of individual head and brain anatomy [19]. But this approach takes long time in two aspects: First, it needs a few hours to build the individual head model. Second, it takes typically several minutes to estimate the E-field, which used to construct and solve a linear system on one location and angle. Hence, it limits its application in situations when rapid adjustments for multiple coil positions and orientations are needed. To overcome the limitations, several computation algorithms [20–22] have been developed to reduce the simulation time.

The deep-learning technique has been recently applied to predict the E-field induced by TMS [21]. This approach is able to significantly reduce the simulation time to much shorter than one second, a significant reduction in prediction time. However, the original framework in [21] has several limitations which compromise the accuracy of the estimated E-field. First, it estimates the magnitude of the E-field without any information about the underlying directions which are useful to investigate the effect of TMS on axonal fiber bundles [23]. Second, the approach in [21] only predicts E-field with the coil placed near a small brain region around the motor cortex with no training and testing data examined for other brain regions. Third, the neural network takes a $T_{1w}$ MRI and the position the TMS coil as input to predict the E-field and neglect the difference between coils. Thus, the trained network is only applicable to predict E-field maps for a specific coil.

In this work, we propose a new deep-learning framework that overcomes the limitations of previous method in [21]. First, our approach predicts the three-dimensional vector E-field instead of its magnitude. Second, our method can be used to predict E-field with the TMS coil placed at different positions over the whole brain. Third, our method uses the change of vector potential, i.e., the dA/dt map, of the TMS coil as input to predict E-field. Thus, the trained DNN can be applied to predict E-field for different types of coils. We have developed four deep neural networks (DNNs) that use different types of imaging data to predict vector E-field. Similar to the method in [21], the first two DNNs were trained based on $T_{1w}$ MRI images to predict E-field simulated using isotropic or anisotropic tissue conductivity tensors, respectively. The other two DNNs take the anisotropic tissue conductivity maps derived from diffusion MRI as the input to predict the E-field maps. By comparing the prediction results of the four DNNs, we can examine if the additional information provided by diffusion MRI can enhance the prediction accuracy. Moreover, the proposed DNNs have a novel network architecture, named 3D-MSResUnet, which integrates the residual module and deep supervision mechanism in a multi-scale way [24, 25] to the standard 3D-Unet architecture [26] to achieve

better performances. The performance of the trained DNNs has been examined using different testing datasets and different types of coils with multiple evaluation metrics.

## Materials and methods

### MRI data and VCM

$T_{1w}$ and diffusion MRI data of 65 randomly selected subjects from the 100 unrelated subjects Human Connectome Project (HCP) database were used in this study [27]. The $T_{1w}$ MPRAGE image was acquired with 0.7 mm isotropic voxels. The diffusion MRI data has 1.25 mm isotropic voxels and three non-zero b-values at b = 1000, 2000, 3000 s/mm$^2$ with 90 gradient directions at each b-shell. The HCP diffusion MRI datasets were preprocessed with corrections for head motion and distortions and were co-registered with $T_{1w}$ MRI [28]. We applied the *mri2-mesh* command from the SimNIBS software (version 3.0.8) [15] which integrates the FreeSurfer toolbox [29] to construct the volume conductor models (VCMs) for each subject using the $T_{1w}$ MRI data. Then, we extracted the b = 0 and 1000 s/mm$^2$ volumes of the diffusion MRI data and applied to the *dwi2cond* command to estimate anisotropic tissue conductivity tensors. Higher b-values were not included in the analysis to ensure that the proposed network is also applicable to diffusion MRI acquired from clinical MRI scanners with standard protocols.

### E-field simulation

We used Matlab (R2015b, Mathworks, Natic, MA) and SimNIBS (version 3.0.8) to simulate E-field maps using the HCP dataset for training and testing the neural networks. Two sets of E-field maps were simulated for each subject by using scalar-valued tissue conductivity and anisotropic conductivity tensors, respectively. The scalar-valued tissue conductivity for white matter, gray matter, CSF, bone and scalp were set as 0.126, 0.275, 0.1654, 0.01 and 0.465 S/m, respectively, which were the default values in SimNIBS. Moreover, the distance from the coil to the scalp was 4 mm which was also the default value. For anisotropic conductivity tensor based simulations, we followed the recommendation of [30] to use volume normalized conductivity tensors to model the anisotropic conductivity in brain tissue with the eigenvectors consistent with those of the diffusion tensor models and with the geometric mean of the eigenvalues identical to the standard isotropic conductivity [31]. In order to generate E-field maps with different coil centers and orientations, we sampled the position of the coil from the positions of the EEG 10–10 system and with the coil handle directed to 78 different directions with approximately 4.6$^o$ angular resolutions. Thus, 52 coil center positions and 78 directions at each position were selected to simulate E-field maps, which provided a total number of 4056 (52*x78)* samples for each subject for either scalar-valued conductivity or anisotropic conductivity based E-field simulations.

### Training and prediction strategies

The results in [21] have shown that a neural network trained by data from 20 subjects can accurately predict E-field simulated using $T_{1w}$ MRI. Accordingly, we selected 20 subjects from the HCP dataset in an initial experiment to train the neural networks. We used two sets of input images to train the neural networks. The first set of training data included the $T_{1w}$ MRI and the temporal change of vector potential map, i.e., the dA/dt map, of the magnetic coil, which provides a 4-dimensional input to the DNNs. This set of input data was applied to train two DNNs to predict E-field distributions simulated based on isotropic and anisotropic conductivity maps, respectively, with trained DNNs being denoted by *T1-iso20* and *T1-aniso20*. The second set of training data included the principal eigenvector from the conductivity tensor

and the dA/dt map of the coil, which provided a 6-dimensional input to the DNN, with the trained DNN being denoted by C-20. This set of data was used to predict anisotropic conductivity tensor-based E-field maps to investigate the additional advantage of using high-dimension data provided by diffusion MRI.

Different from the method in [21], the proposed DNNs not only predict vector E-field maps but also predict it in the whole brain as opposed to scalar E-field maps in a small region. In particular, the dimension of the output data of the proposed DNNs is 180x220x120x3, whereas dimension of the output of the method in [21] is 72x144x24x1. To examine if additional training data was needed to improve the prediction accuracy of the large dimensional output data, we added 40 more subjects to continue the training of the C-20 models, with the result being denoted by *C-60*. Thus, the total number of training datasets for *C-60* was equal to 243,360 (4056x60), whereas the number of datasets used for T1-iso20, T1-aniso20 and C-20 was equal to 81,120 (4056x20).

In summary, four DNNs were trained to predict E-field maps based on isotropic or anisotropic tensor based tissue conductivity. The T1-iso20 model were trained using 20 subjects to predict isotropic conductivity based E-field maps by using $T_{1w}$ MRI and the dA/dt vector field of the TMS coil. The T1-aniso20 model were trained using the same set of input data to predict anisotropic conductivity based E-field maps. The C-20 model was also trained using 20 subjects to predict anisotropic conductivity based E-field maps by using anisotropic conductivity tensors and the dA/dt map. The C-60 model was further trained based on the C-20 model by using additional 40 training subjects.

## Network architecture

The standard network architecture that has been widely used for image-based learning, including [21], is a structure called U-net which consists of a sequence of encoder and decoder modules. In [32], a variant of U-net with the residual module was introduced to solve the exploding or vanishing gradients problem and improve the training efficiency [33]. It was shown that the residual module combined with standard U-net has better performance than the standard U-net in [32, 34]. In addition, [35, 36] proposed a different variant of U-net to pass the features extracted from early stages to later stages of decoders to improve the network prediction performance using multi-scale features.

In this work, we introduce a new network architecture, which is named 3D-MSResUnet, by combining the residual modules introduced in [33] and the approach from [35] to integrate multi-scale features based on U-net architecture. 3D-MSResUnet is a fully convolutional network whose architecture is illustrated in Fig 1. The dark arrows represent the short skip connection of residual modules, and the light arrows mean the short skip connection that pass the feature maps of decoders to the encoders. All cuboids with various colors represent the feature maps extracted from the 3D-MSResUnet. Note that the gray cuboids indicate multi-scale feature maps from the decoders.

## Learning strategy

Our networks were trained by minimizing the mean squared error loss between the predicted E-field and the reference data provided by SimNIBS (version 3.0.8) [15]. The network can be considered as a nonlinear regression model, which was trained to fit the E-field distribution calculated by FEM approach.

We first initialized the network weights using the method proposed in [37]. We then used the RAdam (Rectified Adam) [38] optimizer for network training with modulate parameters $\beta_1 = 0.9$, $\beta_2 = 0.999$ and the initial learning rate $lr = 0.002$. Step learning rate strategy was

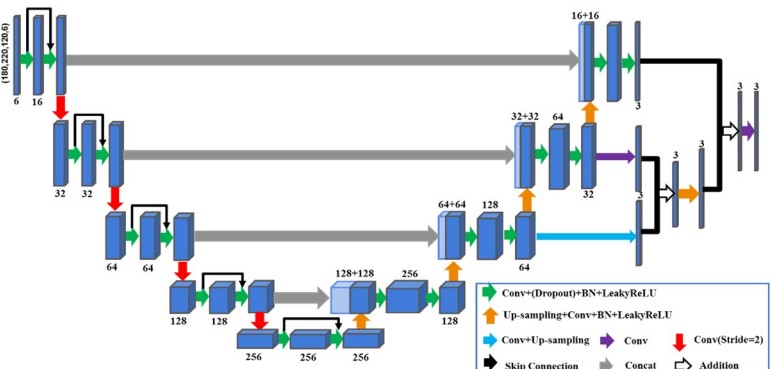

**Fig 1. Illustration of the architect of 3D-MSResUnet.** The last few cuboids shown in dark gray colors represent the mechanism for integrating multi-scale features maps from decoders.

employed with the initial *lr* decayed by gamma = 0.5 after every 5 epochs. The total parameters were iteratively updated using backpropagation for a mini-batch size 4. The parameters were iteratively updated for 25 epochs, where each epoch had $2x10^5$ iterations. The network had 31 convolutional layers totally as illustrated in [Fig 1]. Four 3D-MSResUnet models, i.e., $T_1$-*iso20*, $T_1$-*aniso20*, *C-20* and *C-60*, were trained with the same architecture. The $T_1$-*iso20* and $T_1$-*aniso20* models were trained by using the $T_{1w}$ MRI and the dA/dt maps as input, with training data sampled from 20 subjects, to predict E-field maps simulated using isotropic scalar-valued tissue conductivity and anisotropic conductivity tensors, respectively. The size of the combined input data was 180x220x120x4. The *C-20* and *C-60* models were trained using the principal eigenvector scaled by the corresponding eigenvalues of the conductivity tensor and the dA/dt maps as input, with the training datasets sampled from 20 and 60 subjects, respectively, based on anisotropic conductivity tensors. The size of input data for both *C-20* and *C-60* was 180x220x120x6. The training of *C-60* was initialized by the parameters of the *C-20* model but with additional 40 independent subjects added to the training with $4x10^5$ additional iterations for each of the 25 epochs.

## Testing methods

We conducted the following experiments to examine the performance of the DNN models and their dependence on the type of TMS coils, imaging protocols and the coil positions.

**HCP testing subjects.**   We applied the four trained DNNs to predict E-field maps for 5 independent subjects from the HCP database that were not used in training. Each subject had 4,056 simulated E-field volumes with isotropic and anisotropic tissue conductivity, respectively, using three different coils. The difference between the predicted and simulated E-field maps were compared to examine the performance of the DNNs.

**On the dependence of coils.**   We note that a major difference between the proposed approach and the method in [21] is the inclusion of the dA/dt map as a input to the DNN. Thus, the DNN model can produce different E-field maps for different coils at the same positions. In our experiment, the DNNs were trained based on simulated E-field using a Magstim-70mm-Figure8 [39] TMS coil. To examine the performance for other coils, we applied the C-60 model to predict E-field maps for the MagVenture-MC-B70 coil, which has a similar Figure-of-Eight shape as the trained coil and the Magstim-70mm-Circular coil [40].

**On the dependence of imaging protocols.**   To examine the dependence of the prediction results on imaging protocols, we applied the trained DNNs to predict E-field maps simulated

using the *Ernie* dataset provided by SimNIBS. This dataset was acquired using a standard clinical scanner with 2 mm resolution dMRI data which had much lower resolution compared to the 1.25 mm isotropic voxels in HCP datasets. We simulated E-field maps using both isotropic and anisotropic conductivity maps with three types of coils and compared the results with the predicted data.

## On the dependence of coil positions

To examine if the trained DNNs were able to predict E-field maps with the coil placed at different positions that were not included in the training dataset, we simulated the E-field maps with coil positions sampled at different vertices of the brain surface model at the rostral middle frontal lobe from FreeSurfer [29]. This region contained about 31795 vertices (total) in the 5 testing subjects. E-field maps with anisotropic conductivity were simulated using 3 uniformly sampled orientations at each position, providing a total of 95385 volumes to evaluate the performance of trained neural networks at positions that are different from EEG nodes. The simulation results were compared with the predicted values by the *C-60* model.

## Evaluation metrics

We have evaluated the performance of the neural networks using the following metrics.

**Target overlapping coefficient (TOC).** One main application of E-field simulation is to define the best coil position on the scalp to optimally stimulate a given cortical target with TMS. To evaluate the performance of the predicted E-field for brain targeting, we used two approaches to define the brain target *in the volume space and the surface space*, respectively. In the volume space, we defined the target region as the set of voxels within brain tissue whose magnitude were higher than 95% of all other brain voxels, i.e., the top 5% voxels. To define the target region in brain surface, we first mapped the magnitude of E-field to the brain surface using the *mri_vol2surf* command from FreeSurfer. Then we defined the target region at the set of vertices whose magnitude were higher than *95%* of all vertices. To compare the target regions of the predicted E-field and the reference, we computed their Dice similarity coefficient (DSC) [41], which was named as target overlapping coefficient (TOC). We note that TOC takes value between 0 and 1. Higher TOC implies better similarity between the E-field distributions.

$$TOC = \frac{2TP}{2TP + FP + FN} \tag{1}$$

where the TP, FP and FN mean true positive, false positive and false negative between the prediction E-Field and the reference E-Field.

**E-field peak distance (EPD).** To further evaluate the performance of the predicted E-field maps, we computed the distance between peaks of the predicted and reference E-field magnitude. Similar to TOC, we computed the E-field peak distance (EPD) in both the volume and the surface space, respectively, using the following definition

$$EPD = \|Peak(E_P) - Peak(E_R)\|, \tag{2}$$

where $E_P$ and $E_R$ denote the predicted and reference E-field and *Peak()* obtains the coordinate of the voxel at the gray matter region or the vertex at brain surface with the maximum magnitude. To reduce the influence of outliers on EPD in the volume space, we took the average value of the E-field magnitude in 3x3x3 neighboring voxels within the gray matter region as magnitude of a voxel to determine the location of the peak value of E-field maps for Magstim-70mm-Fig8 and MagVenture-MC-B70 coils. For the less focal E-field distributions

corresponding to the Magstim-70mm-circular coil, the location of the target was determined as the average position of the top 200 voxels with the highest E-field magnitude in the gray-matter region.

**E-field similarity.** We also computed the correlation coefficient between the magnitude of the predicted and reference E-field maps to evaluate their similarities. The E-field similarity score was computed for whole-brain E-field magnitude using the following definition

$$Correlation = \frac{Cov(\|E_P\|, \|E_R\|)}{\sqrt{Var(\|E_P\|)Var(\|E_R\|)}}, \tag{3}$$

where $\|E_P\|$ and $\|E_R\|$ denote the magnitude of the predicted and the reference E-field, respectively.

**Mean absolute error (MAE) and mean relative error (MRE).** The mean absolute error (MAE) was defined as the absolute value of the difference between the predicted E-field magnitude and the reference. The mean relative error (MRE) was defined as the ratio between MAE and the reference E-field magnitude within the corresponding target region. Here, MRE was only computed within the target region to avoid singularity when computing the ratio.

$$MAE = \frac{1}{K}\sum_1^K |\|E_P\| - \|E_R\||, \tag{4}$$

$$MRE = \frac{1}{K}\sum_1^K \left(\frac{|\|E_P\| - \|E_R\||}{\|E_R\|}\right), \tag{5}$$

where K denotes the number of vertices on brain surfaces.

Normalized root-mean-square error (NRMSE): To compare the vector E-field in volume space, we computed the NRMSE measure [42] using the method below

$$NRMSE = \frac{1}{N}\sum_1^N \left(\frac{\|E_P - E_R\|}{\|E_R\|}\right), \tag{6}$$

where N denotes the number of voxels. We computed the NRMSE measure using the whole-brain E-field as well as the target region that contained the 5% voxels around the peak of the reference E-field.

**Mean directional error (MDE).** The direction of E-field vectors was shown to be important to understand the stimulation of white matter in TMS [43]. To this end, we evaluated the angular accuracy of the predicted E-field maps by computing the mean directional error (MDE) between predicted and reference E-field vectors within the reference target region based on the following definition

$$MDE = \frac{1}{N}\sum_1^N acos\left(\frac{E_P.E_R}{\|E_P\|\|E_R\|}\right). \tag{7}$$

Similar to the NRMSE measure, we computed MDE for whole-brain E-field as well as the E-field around the target region.

## Results

### On network training

The neural networks were trained with PyTorch [44] on a Linux workstation equipped with two NVIDIA TITAN RTX GPUs with 48 GB graphics RAM in total. The training time for the *T1-iso20*, *T1-aniso20* and *C-20* models were about 14 days each. The training of the *C-60*

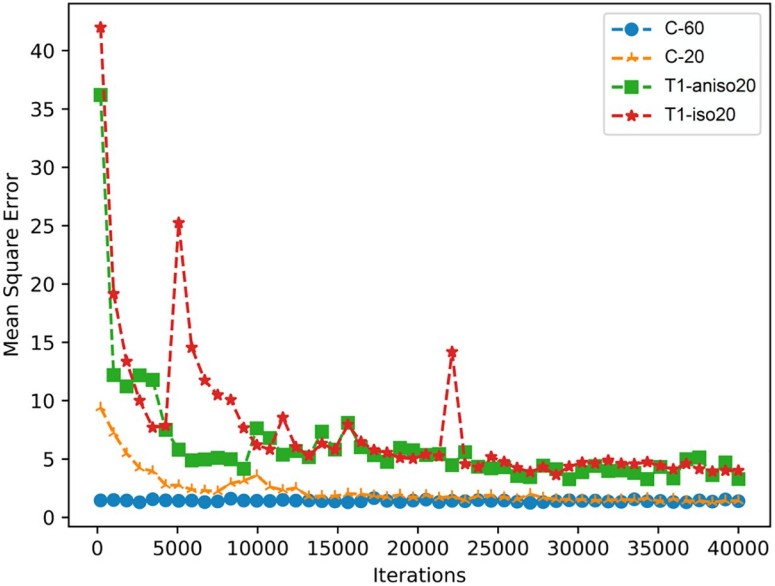

**Fig 2. The training loss for the four DNNs in each iteration.**

model took 28 days. Once the neural networks were trained, they were able to predict an E-field volume in 0.24 s.

Fig 2 shows the mean square error after each epoch in the training stage of the four models. The *C-20* model with anisotropic conductivity tensors in the input had a significantly lower mean square error than the *T1-aniso20* model to predict the same set of E-field maps. The *C-60* model was initialized by the final result of the *C-20* model, which provided a relatively lower mean square error at the beginning of the training. But final mean square errors of the *C-20* and *C-60* models were similar and were lower than the final mean square errors of the T1-iso20 and the T1-aniso20 models.

## Surface-space evaluations

Table 1 shows the evaluation metrics that compare the predicted and the reference E-field maps on brain surfaces. Overall, the T1-iso20 and T1-aniso20 models had similar performances in predicting E-field maps with isotropic and anisotropic conductivity, respectively. The C-20 model overperformed the T1-iso20 model with higher TOC and correlation measures and lower EPD, MAE and MRE measures, indicating that conductivity maps provide better performance than $T_{1w}$ MRI to predict E-field maps. The C-60 model had similar

**Table 1. Target overlapping coefficient (TOC), E-field peak distance (EPD) error [mm], correlation, mean absolute error (MAE) [V/m] and mean relative error (MRE) and mean directional error (MDE) between the VCM and DNNs on gray matter region of the whole brain for T1-iso20, T1-aniso20, C-20 and C-60 and the rostral middle frontal area for C-60 on brain surfaces.**

| Mean (SD) | EEG positions | | | | Non-EEG positions |
|---|---|---|---|---|---|
| | *T1-iso20* | *T1-aniso20* | *C-20* | *C-60* | *C-60* |
| TOC | 0.894(0.045) | 0.906(0.024) | 0.946(0.012) | 0.956(0.009) | 0.962(0.007) |
| EPD (mm) | 3.9591(6.68) | 3.736(6.055) | 1.638(3.951) | 1.394(3.663) | 1.300(3.167) |
| Correlation | 0.983(0.038) | 0.986(0.008) | 0.995(0.004) | 0.997(0.004) | 0.997(0.0015) |
| MAE | 0.018(0.006) | 0.017(0.005) | 0.010(0.002) | 0.008(0.001) | 0.008(0.001) |
| MRE | 0.154(0.059) | 0.152(0.062) | 0.089(0.014) | 0.077(0.013) | 0.077(0.009) |

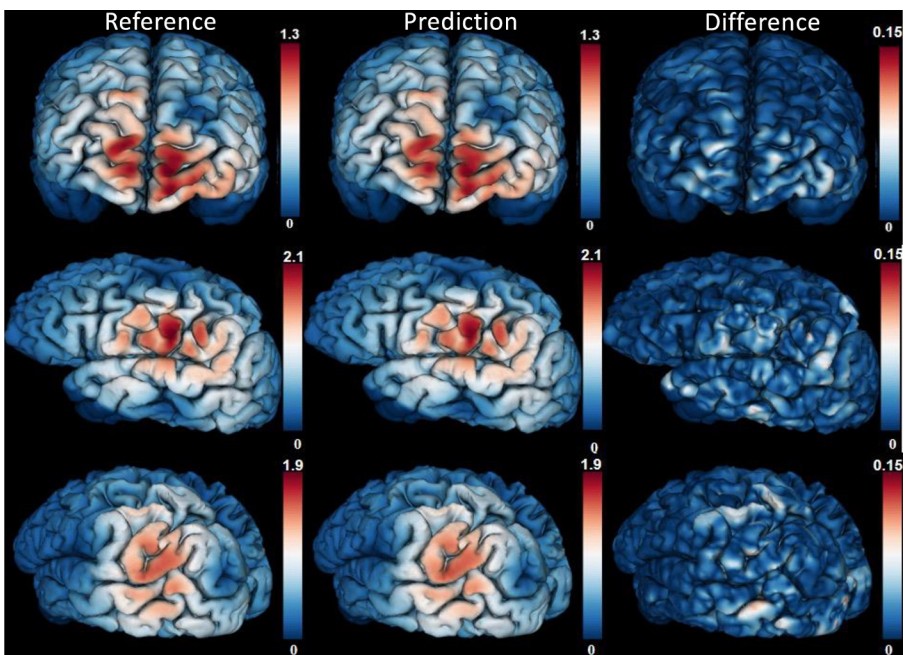

**Fig 3. The magnitude [V/m] of E-field from VCM (the first column) and C-60 (the middle column), and the absolute difference between VCM and C-60 (the last column) with the Magstim-70mm-Fig8 placed at three positions of one testing subject.**

performance as the C-20 indicating that using datasets from 20 subjects can provide reliable training results which is consistent to conclusions in [21]. The performance of the C-60 model at non-EEG positions that were no included in the training dataset was similar to the performance at EEG positions. In particular, the EPD was about 1.3 mm indicating that C-60 provided high accuracy for localizing TMS targets on brain surfaces.

Fig 3 illustrates three samples of the predicted E-field magnitude by the C-60 model (left column), the corresponding ground truth (middle column) and their differences (right column). The coil center of the three E-field maps were at the *Fpz*, *C5* and *P5* EEG positions. Though the MRE was about 7.7% over the entire brain surface, the relative error in some regions was 10% or higher. More detailed comparisons for the E-field maps at non-EEG positions is shown in S4 Fig in S1 File. Moreover, the topographic maps for the evaluation metrics at EEG positions in surface space can be found in S5 Fig in S1 File. A detailed illustration of the evaluation metrics at non-EEG positions at the rostral middle frontal lobe is shown in S5 Fig. Moreover, a video that shows the E-field maps at different non-EEG positions is also provide in the S1 File.

## Volume-space evaluations

Table 2 shows the average target overlapping coefficient (TOC), E-field peak distance (EPD), the correlation measure, the normalized root-mean-square error (NRMSE) and the mean directional error (MDE) of the E-field vectors. The NRMSE and the MDE measures were evaluated for both whole-brain E-field maps and target regions. The second to the fifth columns show the average values of these metrics from all 20280 testing volumes from 5 HCP subjects with coil placed at the same set of positions at in the training data. The last column shows the performance of the C-60 model with the coil placed at a different set of positions.

**Table 2. Performance of the models using independent HCP datasets.**

| | EEG positions | | | | Non-EEG positions |
|---|---|---|---|---|---|
| | *T1-iso20* | *T1-aniso20* | *C-20* | *C-60* | *C-60* |
| TOC | 0.874(0.033) | 0.878(0.027) | 0.905(0.014) | 0.914(0.012) | 0.931(0.008) |
| EPD (mm) | 8.225(8.539) | 8.709(8.789) | 5.588(7.112) | 4.977(6.546) | 3.842(5.671) |
| Correlation | 0.978(0.026) | 0.977(0.008) | 0.984(0.003) | 0.986(0.003) | 0.989(0.001) |
| NRMSE | 0.264(0.0744) | 0.277(0.083) | 0.205(0.019) | 0.187(0.017) | 0.217(0.019) |
| NRMSE (target) | 0.154(0.033) | 0.153(0.031) | 0.107(0.010) | 0.097(0.009) | 0.104(0.007) |
| MDE | 13.455(3.338) | 13.770(4.039) | 9.969(1.133) | 9.117(0.977) | 10.814(0.930) |
| MDE (target) | 6.688(2.545) | 6.605(1.299) | 4.841(0.426) | 4.474(0.368) | 4.957(0.412) |

The evaluation metrics include the target overlapping coefficient (TOC), the E-field peak distance (EPD) error, the Correlation, the Normalized root-mean-square error (NRMSE) and the Mean directional error (MDE) between the VCM and DNNs on gray matter region of the whole brain for T1-iso20, T1-aniso20, C-20 and C-60 and the rostral middle frontal area for C-60 in the volume space.

The T1-iso20 and T1-aniso20 models had similar performances for most evaluation metrics. The C-20 model had better performance than the T1-aniso20 model with increased TOC and correlation and reduced NRMSE and MDE. The NRMSE and MDE for C-20 were both lower than T1-aniso20 at target regions than the average results. The C-60 model had slightly improved performance than C-20 model. The EPD measures for all coils were higher than the surface-based results shown in Table 1. Thus, mapping the predicted E-field from the three-dimensional space to brain surfaces can improve the precision of target localization. The NRMSE and MDE both had relative lower values at target regions compared to the average results over the whole brain. The MDE for C-20 at target regions was about 4.841 degree which was lower than the 6.605-degree error for the T1-aniso20 model, indicating that the anisotropic conductivity tensor provided by diffusion MRI was helpful to improve the angular precision.

The last column of the Table 2 shows that the *C-60* model had similar performance at coil positions that were not included in the training dataset. Thus, the trained networks can potentially be applied to predict E-field with arbitrary coil positions and orientations.

Table 3 shows the evaluation metrics of the C-60 model for predicting the E-field with three types of coils for the HCP data and Ernie dataset. All evaluation metrics for the Magstim-70mm-Fig8 and MagVenture-MC-B70 coils had similar values for both the HCP and Ernie datasets, but the NRMSE and MDE for the Magstim-70mm-Fig8 coil were relative lower. Moreover, the HCP datasets had lower EPD, NRMSE and MDE than the Ernie dataset, indicating the dependence of the DNN on imaging protocols and resolutions. The last column shows the results for the Magstim-70mm-Circular coil. The corresponding EPD was much higher than the result for the other two coils which may be related to the less focal distribution of the E-field.

Fig 4 illustrates the simulated and the predicted E-field maps by the C-60 model for the three types of TMS coils placed at the same position of a testing subject of the HCP dataset. The E-field maps corresponding to the Magstim-70mm-Fig8 and MagVenture-MC-B70 coils have similar distributions around the target regions because the two coils have similar structure and size. But the MagVenture-MC-B70 coil had higher prediction error especially in brain areas outside of the peak regions. Though the Magstim-70mm-Circular coil has a different structure as the Magstim-70mm-Fig8 coil used in DNN training, the C-60 model can still predict similar E-field distributions as the simulated data. But the prediction error is higher than the other two coils.

**Table 3. Performance of the C-60 model using different coils for the HCP and Ernie dataset.**

| | | *Magstim-70mm-Fig8* | *MagVentur-MC-B70* | *Magstim-70mm-Circ* |
|---|---|---|---|---|
| TOC | HCP | 0.914(0.012) | 0.909(0.012) | 0.890(0.012) |
| | Ernie | 0.905(0.011) | 0.895(0.015) | 0.8858(0.014) |
| EPD (mm) | HCP | 4.977 (6.546) | 4.789(6.636) | 6.257(6.174) |
| | Ernie | 7.591(8.588) | 7.236(8.951) | 18.451(12.620) |
| Correlation | HCP | 0.986(0.003) | 0.986(0.003) | 0.985(0.003) |
| | Ernie | 0.982(0.004) | 0.981(0.004) | 0.9806(0.005) |
| NRMSE | HCP | 0.187(0.017) | 0.312(0.045) | 0.244(0.056) |
| | Ernie | 0.3375(0.117) | 0.511(0.194) | 0.1123(0.3503) |
| NRMSE (target) | HCP | 0.097(0.009) | 0.116(0.012) | 0.105(0.016) |
| | Ernie | 0.121(0.016) | 0.156(0.018) | 0.1418(0.019) |
| MDE | HCP | 9.117(0.977) | 11.534 (1.966) | 12.432(3.357) |
| | Ernie | 15.985(4.415) | 18.382 (9.776) | 18.3903(5.973) |
| MDE (target) | HCP | 4.474(0.368) | 4.965(0.494) | 4.407(0.592) |
| | Ernie | 5.474(0.889) | 5.935(1.096) | 5.2675(1.0954) |

The evaluation metrics include the Target overlapping coefficient (TOC), the E-field peak distance (EPD) error, the Correlation, the Normalized root-mean-square error (NRMSE) and the Mean directional error (MDE) in the volume space.

## Topographical illustrations

Fig 5 shows the topographic maps of the evaluation metrics associated with different coil positions. At each EEG position, the metric was the average value from 390 samples (5 subject and 78 directions). These topographic maps illustrate the dependence of the performance of network models on coil positions. In particular, the target overlapping coefficient (TOC) was symmetrical between left and right head. The larger E-field peak distance (EPD) and MAE appeared in the front of head. The correlation was more or less uniform in all brain regions except for the midline. It can be seen that the performance calculated by different metric on

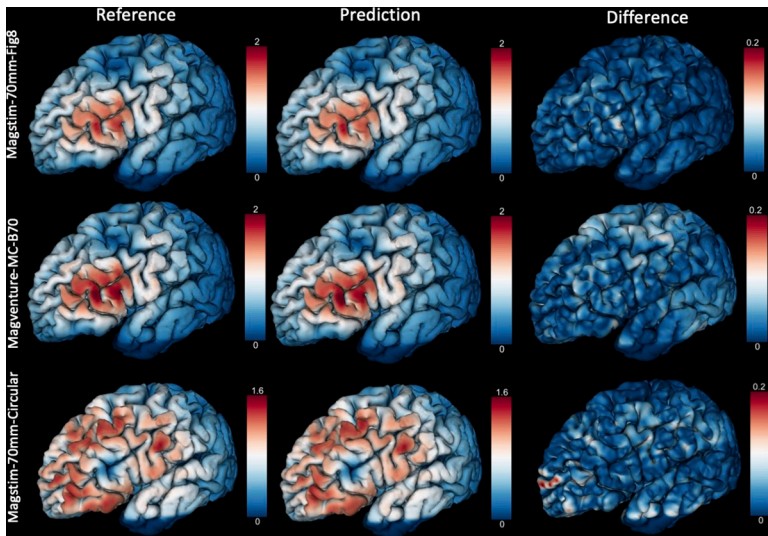

**Fig 4. The magnitude [V/m] of E-field from VCM (the first column) and C-60 (the middle column), and the absolute difference between VCM and C-60 (the last column) for three types of TMS coils placed at the same position of a testing subject of the HCP dataset.**

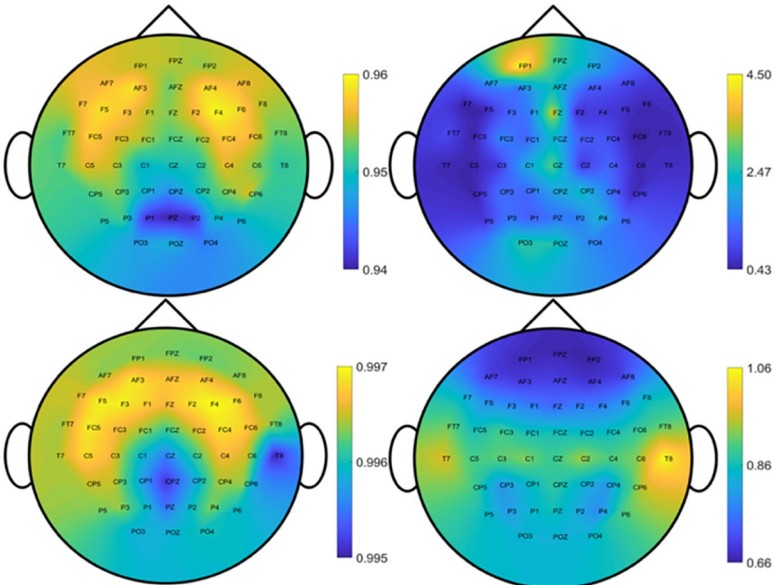

**Fig 5. The distribution of TOC, EPD, Correlation and MAE [mV/m] on EEG position with C-60 model on the surface space.**

EEG positions. We noted that it has nearly symmetry between left head and right head, especially for TOC and MAE. Moreover, the performance is different on different EEG positions.

## Discussion and conclusions

In this work, we have introduced a deep-learning based framework for E-field prediction for TMS applications. We examined its performance using different types of training data, coils and imaging protocols. Our method has several major differences compared to the DNN method in [21]. First, our method uses the dA/dt map of the magnetic coil to predict vector E-field. Thus, the trained DNN model can be applied to predict E-field maps for different TMS coils. Second, we proposed a novel architecture of neural networks, i.e., 3D-MSResUnet, to improve the prediction accuracy by combination of residual module and multi-scale feature maps into 3D-U-net architecture. Third, our method used anisotropic conductivity maps to improve the prediction accuracy of E-field maps. Several key results and limitations are summarized below:

### Conductivity map vs $T_{1w}$ MRI

The experimental results showed that the T1-aniso20 model can use $T_{1w}$ MRI to predict E-field maps based on anisotropic conductivity with similar performance as the T1-iso20 model for predicting E-field based on isotropic models. While using $T_{1w}$ MRI to predict E-field based on anisotropic models is an advantage compared to physics-based FEM method, the prediction accuracy is worse than the results of C-20 that uses anisotropic conductivity tensors for E-field prediction. On the other hand, the dependence the C-20 and C-60 models on diffusion tensors limits their applications in situations when diffusion MRI is not available.

### On the dependence on positions, coils and imaging protocols

The experimental results showed that the trained DNN can predict E-field maps for coil positions that were not included in the training dataset. The whole-brain feasibility of the trained

networks was attributes to the extensive training of using 243,360 data volumes which contained information about heterogeneity in brain anatomy and differences in coil positions and orientations. Moreover, the results have also shown that the performance of DNN models depend on both the shape of coils and imaging protocols. Low-resolution MRI images can reduce the targeting accuracy for the DNN models. The results for the MagVenture-MC-B70 and Magstim-70mm-Fig8 coils were similar since both coils have similar shapes. But the performance significantly degraded for the Magstim-70mm-circular coil. Thus, further training and different training strategies are needed to improve the performance for different coils and imaging protocols.

## On localization accuracy

Several evaluation metrics were applied to examine the prediction accuracy of the DNN models. The experimental results showed that the target overlap ratio of the C-60 model on brain surfaces was 95.6% and the E-field peak distance was 1.3 mm and the correlation coefficient was about 0.997. The distance error increased in the volume space to about 11.3 mm. The T1-iso20 and T1-aniso20 models had reduced targeting accuracy with the E-field peak distance on brain surfaces being 3.959 mm 3.736 mm, respectively, and 8.225 mm, 12.601 mm in volume spaces. The angular error of the C-20 and C-60 models in target regions were about 4.841 and 4.474 degrees which were lower compared to the 6.688 degree for T1-iso20 and 6.605 degrees for T1-aniso20. But the angular errors were much higher than the fast quadrature method [22]. Thus, a different training strategy may be needed to improve the angular precision of the DNN models, which will be explored in our future work.

## On prediction speed

The prediction of a whole-brain E-field volume using the trained neural networks took about 0.24 s. In practice, additional time is needed to apply rigid transformation to the dA/dt map according to the coil position which is expected to take much shorter computation time. Moreover, the prediction speed is still slower than the fast quadrature method [22] and the DNN based method [21] though the predicted E-field by these methods is only in a smaller region of the brain. More recently, a fast computational algorithm was introduced in [45] to estimate E-field in a selected ROI so that the E-fields generated by coils placed at 5900 different scalp positions and 360 orientation per position can be computed under 15 minutes. In [46], a rapid algorithm was introduced to compute E-field in ~100 ms on a cortical surface mesh with 120k facets and with about 5 hours of preparation time. Compared to these methods, the merits of our DNN-based method lie in the simplification in data preprocessing since it does not need mesh models and the acceleration in whole-brain E-field volume prediction. But significant improvements are needed to accelerate the prediction of E-field in target ROI to optimize coil positions and to achieve real-time prediction on brain surfaces. For this purpose, we will improve the architecture and training approach of the DNN in our future work to directly predict E-field in a selected ROI or on brain surfaces. We expect that reduced data dimension and simplified network architecture can significantly reduce the prediction time.

Finally, we note that the DNN methods have several limitations. First, the neural networks were trained to predict E-field simulated with fixed values for tissue conductivity, i.e., specific values of conductivity for different tissue types, a specific coil and specific imaging protocols, which are limitations compared to physics-based FEM algorithms. Second, the trained DNN models not only depend on the type of coils, imaging protocols but also the data processing methods. In particular, tissue segmentations in this study were obtained based on $T_{1w}$ MRI, but more accurate results can be obtained by using both $T_{1w}$ and $T_{2w}$ MRI. Thus, further

development and training of the DNN models are needed to integrate different tissue segmentation approaches for more accurate prediction results. Third, the DNN models were only trained based on data from health subjects whereas physics-based FEM algorithms have more broad applications for patients with tumors or lesions. Thus, the goal of the DNN-based method is not to replace FEM approaches. But the DNN-based methods can potentially be useful to accelerate the prediction of E-field in situations when their performances are validated. It is also a potentially useful tool to use only anatomical images, e.g., $T_{1w}$ MRI, to predict E-field based on anisotropic conductivity tensors when diffusion MRI is not available in clinical settings, though further improvements in prediction accuracy are needed.

## Supporting information

**S1 File.**
(PDF)

**S1 Video.**
(AVI)

## Acknowledgments

This study was partially completed while XG visited the Psychiatry Neuroimaging Laboratory at Brigham and Women's Hospital in 2019. The authors would like to thank Fan Zhang and Joshua Goldenberg for their helpful assistance.

## Author Contributions

**Conceptualization:** Yogesh Rathi, Joan A. Camprodon, Lipeng Ning.

**Data curation:** Lipeng Ning.

**Formal analysis:** Guoping Xu.

**Funding acquisition:** Lipeng Ning.

**Investigation:** Guoping Xu, Lipeng Ning.

**Methodology:** Guoping Xu.

**Supervision:** Lipeng Ning.

**Writing – original draft:** Guoping Xu, Lipeng Ning.

**Writing – review & editing:** Guoping Xu, Yogesh Rathi, Joan A. Camprodon, Hanqiang Cao, Lipeng Ning.

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
