## [Decision Letter · Decision Letter 0]

23 Dec 2020

PONE-D-20-30151

Toward real-time electric field mapping in transcranial magnetic stimulation using deep learning

PLOS ONE

Dear Dr. Ning,

Thank you for submitting your manuscript to PLOS ONE. After careful consideration, we feel that it has merit but does not fully meet PLOS ONE’s publication criteria as it currently stands. Therefore, we invite you to submit a revised version of the manuscript that addresses the points raised during the review process.  In particular, both reviewers expressed significant concerns about the evaluation of the proposed approach, and a lack of comparison to competing methods.   Both reviewers also raised questions about the methodology.  Reviewer 1 further suggests a potential reframing of the primary results of the paper.  

We look forward to receiving your revised manuscript.

Kind regards,

Dzung Pham

Academic Editor

PLOS ONE

Journal Requirements:

2. Thank you for providing links to your data and code in your cover letter. We ask that you provide this information (" The data used in this paper are publicly available from the Human Connectome Project. The Python script used in this paper is available at Github via " ext-link-type="uri" xlink:type="simple">https://github.com/LipengNing/Efield.git."), along with a link to the Human Connectome Project in both your Data Availability Statement and your Methods section.

Reviewers' comments:

Reviewer's Responses to Questions

**Comments to the Author**

1. Is the manuscript technically sound, and do the data support the conclusions?

Reviewer #1: No

Reviewer #2: Partly

2. Has the statistical analysis been performed appropriately and rigorously? 

Reviewer #1: N/A

Reviewer #2: Yes

3. Have the authors made all data underlying the findings in their manuscript fully available?

Reviewer #1: No

Reviewer #2: Yes

4. Is the manuscript presented in an intelligible fashion and written in standard English?

Reviewer #1: Yes

Reviewer #2: Yes

5. Review Comments to the Author

Reviewer #1: The manuscript left me with mixed feeling. It contains a good idea, but in its current state, it is unsuitable for publication.

First, the bad news: The authors use a DNN for approximate computation of a problem with known physics-based solution. They do not, however, compare their results to existing fast computation methods, but rather provide their ‘accuracy’ without any reference or comparison with either existing methods or requirements of the application the authors suggest the method to be used. The approach of using DNN to replace physics-based models is also questionable, see point 30 which paraphrases authors own critique of the earlier DNN work by (Yokota et al., 2019).

Then, the good news: their result with ‘T1-20’ model is interesting. This is an application where the DNN-based approach may shine. They did reach almost full accuracy of their approximate solver (with full input data) with limited input data. This should be the main result: with DNN, we can estimate the induced electric field of an anisotropic VCM without having to measure the anisotropy (which is expensive for subject-specific models). However, even with this change, the authors must perform additional comparisons. The accuracy of approximate DNN model must be compared to that of isotropic physics-based model. See my point 31. This alone is a reason I will currently have to suggest rejection, but most certainly with an option to resubmit.

Further, to ensure that the model is really robust to change of coil, the tested coil model should be fundamentally different from the teaching coil model. Now they are two figure-of-eight coils with fairly similar sizes (one step down in size). The testing coil could be, e.g., a circular coil or an H-coil.

Finally, it seems that the authors have used the SimNIBS with incorrect parameters (omit the T2 images, which is needed to correctly predict the thickness of the skull in the modelling pipeline). Thus, much of the results need to be re-computed.

-

I will begin with minor technical observations:

The authors, for no apparent reason, limit availability of their data. Their data availability statement has arbitrary requirement of “The data that support the findings of this study are available from the corresponding author upon *reasonable* request.” The authors provide no legal or ethical reason to limit access in this manner.

I further wonder why this work is submitted to PLOS ONE and not into some of the more specialized journals on neurostimulation methods.

-

Then, my other observations are in the order of their occurrence.

The abstract:

1. First sentence: TMS, in addition to being a ‘neuromodulation’ method is a ‘neurostimulation’ method. The latter is more descriptive, as neuromodulation is possible also with the subthreshold stimulation methods.

2. The second sentence is misleading: “Due to the complex structure of the brain and the electrical conductivity variation across different tissues, it is difficult to exactly identify the brain region stimulated by TMS, which is important to improve the treatment efficacy and understand the underlying mechanism.”

Even based on this work, this computation is very straightforward with readily available toolkits to compute this. Further, this work, at best tries to reproduce the accuracy of these existing methods. Hence, calling this problem difficult is misleading.

3. The “relative long computation time” is an incorrect claim. Realistic VCM models have been solved in under 25 ms, a further factor of 10 faster than the approximate solver shown in this work. And, compared to fast FEM implementations (such as the 3-second computation in Yokota et al., 2019) the authors speedup does not make a fundamental difference.

This is because the authors fail to define what they mean with “real-time computation”. Their computation speed is much lower than in any of the previous three methods for ‘real-time computations’ (the local spherical models, the previous magnitude-only isotropic-VCM DNN, or a hardware-accelerated, fast-quadrature, isotropic-5C-VCM BEM).

The authors current computation speed of 240 ms = 4.2 computations per second = 4.2 frames per second is still too slow for neuronavigation (where the operator needs a smooth feedback, at minimum at least 15 frames per second, i.e. below 67 ms, and ideally 30 or 60 frames per second, i.e., 33 ms and 17 ms, respectively).

The spherical model rans readily at these speeds, as did the earlier DNN which computed the fields in 10–30 ms (Yokota et al., 2019), or the real-time BEM which computed the fields in 15–23 ms based on it computing 46–65 coil positions per second (Stenroos Koponen, 2019).

In the current performance bracket, the model does not provide a quantitative improvement over the fast FEM solvers (about 3 s, not the number suggested in the work at 30-60 s), but is quantitatively too slow for the suggested ‘real-time’ application by a factor of at least 4.

4. Abbreviation FEM (finite element method) should be defined before its first use in the abstract.

Introduction:

5. Line 42: the TMS coil current is pulsed, not oscillating (which suggests a continuous stimulation instead of discrete pulses). Even at highest repetition rates, the duty cycle of TMS is less than 1%, and typically it is much less than 0.1%.

6. Line 52: the “multi-sphere method” is commonly known as “local sphere model”. This is also the case for the provided reference by the authors. Further, instead of the text-book chapter, the local spherical model should probably be attributed to its source, based on the book (Ilmoniemi et al., 1996), or as that work, similarly to the cited text-book chapter, is not readily available, their journal publication (Ilmoniemi et al., 1999).

7. Line 53: TMS-specific BEM should be attributed to more correct, older references such as (Salinas et al., 2009) or (Nummenmaa et al., 2013).

8. Line 54: TMS-specific FEM, similarly to above, should be attributed correctly (Miranda et al., 2003).

9. Line 54: remove word “sophisticated”, as it does not apply only to FEM VCM. The BEM VCM are equally, or sometimes even more sophisticated given the recent advanced in fast-multipole methods.

10. Further, related to the claim “but also anisotropic tissue conductivity, which cannot be done by other methods” (line 56), the authors glossed over FDM methods which can model the anisotropy (e.g., De Geeter et al., 2011). The authors further failed to justify why the anisotropy in particular is of such a high importance (given the uncertainty conductivity values, for both isotropic and anisotropic case).

11. Line 66: BEM is not an accelerating algorithm!

12. Further, the sentence in line 66 should contain both (Stenroos Koponen, 2019) and (Yokota et al., 2019), the existing real-time solvers.

13. Finally, I do not see any fundamental reason why (Laakso Hirata, 2012) could not include also the anisotropy. It is an FEM after all. They likely followed the common convention in TMS-stimulation literature of omitting the anisotropy as we do not really know the anisotropic conductivity tensor. (As the authors also admit when they slightly latter use one of the three rules of thumb, none of which is even based on the physics, of deriving the anisotropic conductivity values from the DTI data and the not-that-accurately-known isotropic conductivity value.)

14. Line 74: The authors again claim their method capable of real-time simulation in clinical and research setting despite it being too slow for the suggested real-time neuronavigation use.

15. Line 85: Overselling, every existing physics-based computation method (spherical models, FEMs, FDMs, BEMs) predicts the three-dimensional vector E-field: “In particular, our approach predicts the three-dimensional vector E-field instead of its magnitude.” The sole method that does not, is the previous DNN method by (Yokata et al., 2019).

16. Line 88: Advertising a weakness of the method as its strength. The ‘C-20’ and ‘C-60’ require much more time-consuming and costly DTI dataset of each individual TMS subject, compared to just T1 (and in most cases T2 for best quality) MRI. It is not a strength to require the extraordinarily high-resolution input files of the HCP, but a limitation: “Furthermore, both diffusion-MRI based conductivity tensors and T1w MRI, which integrate anisotropic conductivity of brain tissues, are used to predict the E-field.”

17. Line 90: Overselling similarly to point 15. All but the previous DNN method allow straightforward change of the coil model.

18. Lines 106–116: there are results at the end of introduction?

The methods or metrics such as “target overlapping coefficient” are not defined here!

This paragraph is entirely impossible to read for a person who has not read the rest of the work before returning here. It should be removed!

Oh, one final thing on this part: the 9-degree error is NOT much lower than a 13-degree error. They are about equal, and both are HORRIBLY BAD VALUES for such a simple quantity that can be modelled with physics-based models instead of DNN.

13-degree error is comparable to a one-shell model, and 9-degree error to omitting the whole of white matter altogether (Stenroos Koponen, 2019). This means that the “approximate DNN model with anisotropy” performs worse than much simpler and faster conventional physics-based models. This is despite requiring much more input data!

Considering this performance, or lack of thereof, the authors use FAR TOO BOLD words such as “the rich information” (line 114) and “ultra-short computational time” (line 114).

Further, if the computation time is from the extraordinarily expensive high-end GPU, it should not be used to describe the suitability to “clinical setting” (line 115).

PS. It is computation time, not computational time. This error repeats throughout the work.

Material and methods (at this point, I will shorten my observations, as I have written a tad overly long response already).

19. Line 124: define “the minimal processing pipeline” (reference or description)

20. Line 125: define “the native space” (the native space of what, a description)

21. Line 126: questionable use of SimNIBS. The VCM construction should be given both T1 and T2 images for better performance.

22. Line 126: Which SimNIBS version it was, and why there is no reference?

23. Line 128: Unnecessary sentence “We note that another command headreco can also be used to generate head models which may lead to different simulation results [Nielsen et al. 2018].”

24. Instead of unnecessary sentences like in point 23, the previous sentences should contain all relevant modelling parameters. (The work further glosses over the entire problem with selection of conductivity parameters, despite being about anisotropic conductivity, which is finicky to the choice of the isotropic baseline values.)

25. Line 137: You seem to provide exact details on version of MATLAB, but for some reason omit them from SimNIBS at all points.

26. Line 146: what does this sentence about the “average degree between the nearest handle directions being about 4.6 degrees” even mean?

27. Line 165: grossly incorrect statement. Just having the T1 image (or even the correct set for good VCM, i.e., T1 and T2) has nothing to do with not computing the direction of the E-field! Is this the reason for the ridiculous claim of point 15? The anisotropy has a (small compared to just the much larger isotropic conductivity differences) effect on the field directions. It is not a fundamental reason for the field directions. I refer now to sentence: “We note that the T1w MRI was used in [Yokota et al., 2019] to predict only the magnitude of the E-field since T1w MRI does not contain diffusion information about the axon orientation.”

28. Line 169: “field” not “filed”.

29. Line 169: I assume you refer to the spatial distribution of the change of the vector potential (dA/dt) at the beginning of the TMS pulse, and not peak temporal changes!

30. Line 177-179: The authors critique of (Yokota et al., 2019) could also be said about their own work! Let me paraphrase: “Thus, the DNN in (THE AUTHORS) is not a solver for the forward model but also needs to learn the well-known underlying physics of electromagnetic induction, which may reduce the prediction accuracy.”

This is basically my main critique of both this work and the work of (Yokota et al., 2019).

In their current state, both methods are FAR LESS accurate than any PHYSICS-BASED solvers. And, in the case of the present manuscript, they are also slower than properly implemented fast physics-based solvers. The approximation by (Yokota et al., 2019) was at least momentarily the fastest approach with more complex than spherical geometry, before being caught up and surpassed by the fast BEM solvers.

31. But, then the positive. Line 194: you actually, should the result survive through the proper cross-validation against the error for a SimNIBS model without the DTI data, try to make things that are not possible with a physics-based model.

The ‘T1-20’ model approximates the DTI dataset without needing one.

This result, however, will need to be compared to the computation against a physics-based model without the DTI data. As, if the physics-based model has prediction errors smaller than 13 degrees, then it is both faster more accurate than the proposed DNN-based model. Such comparison is missing, and without it, I will suggest rejection.

The approximation of the DTI data is only true if the ‘T1-20’ can perform better than just using a physics-based solver without the data. (For which the error is likely much less than 10 degrees.)

32. Line 231-233, rewrite this sentence to make sense of it.

33. Line 282: on scalp, not on skull.

34. Line 293: between E-field distributions, not between the brain regions.

35. Line 303: why is the word reference in quotation marks? Remove them.

36. Line 306-310: This error metric is incorrect, and underestimates the relevant prediction error. The correct method is to compute the magnitude of the vector difference (similarly to previous works with physics-based models).

Your method gives an error of 0 even if you would have predicted the field direction incorrectly by 180 degrees. (See the point about testing the coil with a different type of TMS coil, such as circular coil.)

37. Line 324: Comparing apples to oranges. The computation speed for the DNN is told with two flagship GPUs, and the computation speed for the (not exactly optimized for speed) FEM is told with an undisclosed hardware (probably no GPU acceleration).

38. Line 327: Repeating the incorrect claim that 240 ms computation is ‘real time”. The authors never mention what they actually mean with that. They give example of neuronavigation, for which such computation speed is too slow.

39. Line 339: Significantly as in statistical significance? Or, some other significance? If using the latter used criteria of visual indistinguishability, both are equally good, or rather bad.

Further, line 342: I saw not even a statistically significant difference between the results of C-20 and C-60?

40. Table is on line 345: Why is the angle error omitted from this table?

41. Line 354: what exactly do you mean with “almost visually indistinguishable”?

42. Line 362, figure 3: Is this the difference of fields or difference of magnitude of fields? If former, there is 10% errors in the fields? If latter, well…

43. Line 382: The angle errors are humongous. They are comparable to far simpler head models, not to some simple approximation. Compared to the approximation errors due to the fast quadrature of (Stenroos and Koponen, 2020), where the angle error was 0.1 degrees, the 9 degree and the 13 degree errors are unusable.

44. Line 393, table 2: the location error is also 10 mm even for the ‘C-60’. Such larger error is unusable for neuronavigation!

Also, based on this table, is the angle error rather not 14 degrees and not 13 degrees?

45. Line 407, table 3: the EPD, the correlation error, and the MDE are all unusably large! Such errors seem about the same as the error of a local sphere model? Compare to results of (Stenroos Koponen, 2019).

Discussion and conclusions:

46. Line 430: “novel advantages” what does this even mean?

47. Line 431: this is not a new feature: “First, it can be applied to predict E-field vector volumes with arbitrary coil positions and orientations”

48. Line 433: with angle error close to 10 degrees, I would not exactly call this accurate: “to accurately predict the magnitude and the direction of E-field.”

49. And, then line 439, compare to point 31. Basically, you show that your DNN is worse than a physics-based model, and not fundamentally faster. “Our work shows that deep learning can be further used as a solver for the forward model for estimating whole-brain E-field in TMS.”

I would like to point out that you have far more than 10 times more computational power than the fast-multigrid method by (Laakso and Hirata, 2012), which took just 3 s in (Yokota et al., 2019).

And, you do not even compare to such fast solver (~3 s), but rather compare your GPU-accelerated computation time to a not-optimized for real-time FEM without any GPU acceleration (30-60 s).

50. Line 444-448: No you did not show this. What was your reference model compared to which you showed that the FEM was accurate in its prediction? Did you test for the sensitivity of your solution to your input parameters (which you failed to disclose), or even just the model resolution?

51. Line 467: 9 degree error in volume is not similar in direction. It is horribly badly off.

52. Line 470: 13.7 degrees is not “as low as” but rather horribly bad prediction.

53. Line 473: how is this work applicable to clinical setting? Where do we get all the DTI data, and the computational resources?

54. Line 478: the computation time (not computational time) is not ultra-short.

55. Line 480: visualization in real-time might indeed be useful (not quite essential), but your work fails to deliver such speeds, unlike previous ‘real-time methods’

56. Line 483: Probably soon tenth repetition of this same line does not make it true. Please, do not oversell your work.

57. Line 487: Fundamental problem with understanding physics-based models. A FEM and a BEM will produce practically identical E-field predictions, e.g., (Gomez et al., 2020), and thus the DNN should not change based on which physics-based model it is trained on. This is the whole reason for physics-based models. We get the SAME result with a BEM as with a FEM, or an FDM (given adequate resolution for each).

This is why DNN is also inherently problematic for this type of problem. You essentially try to make it learn very well-known physics, instead of actually just using said physics to compute the result.

58. Line 490: There were other far more important fixed parameters than distance to scalp (which your dA/dt –based approach should in any case be fairly robust on). But then again, this revisits points 31, 57, etc.

DNN is fundamentally wrong tool for solving the forward problem. (The ‘T1-20’ to predict the DTI-FEM result is interesting, however. That is, if you compare it to no-DTI FEM to see if it can actually perform better.)

59. Line 490: for example, you now fix the conductivity parameters. And, of course you model only works for ‘normal’ brains. This makes the method unsuitable for, e.g., neuronavigation of persons with lesions etc. (which is where the physics-based models excel).

60. Line 490: based on what data is the DNN expected to be robust against the change of the long list of parameters you fail to mention at any point in the manuscript?

Reviewer #2: This paper introduces a novel U-net architecture for realtime Efield analysis of TMS. The manuscript is well written and their methods technically sound. However, some important issues need to be addressed:

1) Since they are claiming this is accurate they should mathematically define all of their error metrics.

2) The authors do not compare to standard error metrics like L2, pointwise error like in https://doi.org/10.1016/j.brs.2019.09.015 . Also, others like RDM. What is the relative performance to existing methods.

3) How long does it take to compute dA/dt?;this should be included in computation time.

4) This method has not been trained with different coils, how do we know the trained network will generalize to other coils?

5) In a sense when you move the coil the dA/dt is moved; this is equivalent to moving the brain. So their claim that passing these maps changes something needs a deeper rationale.

6) is the DNN computing both primary plus secondary E-field or just secondary?

6. PLOS authors have the option to publish the peer review history of their article (what does this mean?). If published, this will include your full peer review and any attached files.

Reviewer #1: No

Reviewer #2: No

---

## [Author Response · Author response to Decision Letter 0]

20 Apr 2021

Please find our response letter to the reviewers in the attached files.

---

## [Decision Letter · Decision Letter 1]

9 Jun 2021

PONE-D-20-30151R1

Rapid whole-brain electric field mapping in transcranial magnetic stimulation using deep learning

PLOS ONE

Dear Dr. Ning,

Thank you for submitting your manuscript to PLOS ONE. After careful consideration, we feel that it has merit but does not fully meet PLOS ONE’s publication criteria as it currently stands. Therefore, we invite you to submit a revised version of the manuscript that addresses the points raised during the review process.  Overall, there remains some difference of opinion among the reviewers on the value of this contribution and the use of deep neural networks (DNNs) for E-field modeling.  In particular, as FEM/BEM models continue to improve in speed, it is not clear what niche is filled by DNN approaches.  In addition to responding to reviewer comments, I suggest that the following revisions be made to the manuscript:

Please cite the references provided by Reviewer 4 and briefly discuss contributions of the proposed method compared to these references.Please add a brief discussion on the future of DNN approaches against accelerated FEM/BEM models for E-field modeling.  

If applicable, we recommend that you deposit your laboratory protocols in protocols.io to enhance the reproducibility of your results. Protocols.io assigns your protocol its own identifier (DOI) so that it can be cited independently in the future. For instructions see: http://journals.plos.org/plosone/s/submission-guidelines#loc-laboratory-protocols. Additionally, PLOS ONE offers an option for publishing peer-reviewed Lab Protocol articles, which describe protocols hosted on protocols.io. Read more information on sharing protocols at https://plos.org/protocols?utm_medium=editorial-emailutm_source=authorlettersutm_campaign=protocols.

We look forward to receiving your revised manuscript.

Kind regards,

Dzung Pham

Academic Editor

PLOS ONE

Journal Requirements:

Reviewers' comments:

Reviewer's Responses to Questions

**Comments to the Author**

1. If the authors have adequately addressed your comments raised in a previous round of review and you feel that this manuscript is now acceptable for publication, you may indicate that here to bypass the “Comments to the Author” section, enter your conflict of interest statement in the “Confidential to Editor” section, and submit your "Accept" recommendation.

Reviewer #2: All comments have been addressed

Reviewer #3: All comments have been addressed

Reviewer #4: (No Response)

2. Is the manuscript technically sound, and do the data support the conclusions?

Reviewer #2: Yes

Reviewer #3: Yes

Reviewer #4: No

3. Has the statistical analysis been performed appropriately and rigorously? 

Reviewer #2: Yes

Reviewer #3: Yes

Reviewer #4: Yes

4. Have the authors made all data underlying the findings in their manuscript fully available?

Reviewer #2: Yes

Reviewer #3: Yes

Reviewer #4: Yes

5. Is the manuscript presented in an intelligible fashion and written in standard English?

Reviewer #2: Yes

Reviewer #3: Yes

Reviewer #4: Yes

6. Review Comments to the Author

Reviewer #2: (No Response)

Reviewer #3: In this study, authors propose a method to estimate the induced E-field from the magnetic field generated by TMS coil and the MRI image, using the framework of deep learning. Since the E-field can be estimated by the forward calculation of DNN without solving the problem, the estimation can be significantly accelerated than the FEM. In the experiment, the proposed DNN is evaluated by several metrics, and the effectiveness of the proposed method is confirmed.

To my understanding, the most relevant paper to this study is [21] by Yokota et al at 2019. In terms of TMS-induced E-field estimation using DNN, this study can be considered as a kind of next step of original one [21]. The main new contributions are three folds: (1) estimation of a whole brain region, (2) vector field estimation, and (3) magnetic filed representation of TMS coil parameters.

Contributions (1) and (2) are relatively weak because both vector and whole brain extensions are straightforward by introducing some powerful computational resources. On the contrary, since weak E-fields is often majority in whole brain, the necessity of whole region estimation is unclear for TMS targeting. Moreover, the requirement of the large DNN model with powerful computation (i.e., large memory of GPU) reduces practical usefulness. That is always trade-off.

The contribution (3) is important and it make the paper acceptable in my opinion. By introducing magnetic field representation of TMS coil parameter, the DNN can be trained and applied for various types of coils. It is nice that some computational experiments have conducted by using three different coils. However, it is necessary to care that the overfitting is more easily cased in the proposed framework becasue input space is quite larger than original one [21].

Reviewer #4: Comments on the paper “Rapid whole-brain electric field mapping in transcranial magnetic stimulation using deep learning”

This is a modeling study which aims to use a neural network (DNN) to predict the TMS fields in subjects in a real time. It essentially develops the idea of Ref. [21] with an extra inclusion of the primary TMS field as an input. 65 Connectome subjects and the SimNIBS FEM solver have been used to perform numerical experiments.

While the original idea is interesting, the reviewer still tends to agree with the previous review #1 saying that “The DNN is inherently problematic for this type of problem”.

The first reason is that it is obviously hardly possible to replicate a unique gyral pattern of a subject which, in its turn, uniquely defines the resulting TMS field, based on limited numbers of computations for other subjects (20 or 60 in the paper). Even for those numbers, training the DNN takes weeks, according to the paper, and requires a specially equipped workstation.

The second reason is that, according to the manuscript, after the several weeks (!) of training, the approximate TMS field could be predicted in 0.24 sec. However, the paper does not compare this finding to the performance of the modern physical solvers, which estimate the TMS fields precisely.

As a first example, a new precise physical approach of Gomez et al

[Gomez LJ, Dannhauer M, Peterchev AV. Fast computational optimization of TMS coil placement for individualized electric field targeting. NeuroImage. 2021 Mar;228:117696. doi: 10.1016/j.neuroimage.2020.117696.] allows us to predict the E-fields generated in an MRI-derived head model when the coil is placed at 5900 different scalp positions and 360 coil orientations per position (over 2.1 million unique configurations) under 15 min on a standard laptop computer. This performance of the exact FEM physical solver simply cannot be matched to the approximate results reported by the authors.

As a second example, another new physical approach of Daneshzand et al

[Daneshzand M, Makarov SN, de Lara LIN, Guerin B, McNab J, Rosen BR, Hämäläinen MS, Raij T, Nummenmaa A. Rapid computation of TMS-induced E-fields using a dipole-based magnetic stimulation profile approach. Neuroimage. 2021 Apr 30:118097. doi: 10.1016/j.neuroimage.2021.118097.] allows computation of the E-field in ∼100 ms given ~5 hours of preprocessing time. This performance of the exact BEM physical solver cannot again be matched to the approximate results reported by the authors.

Compared to these studies, the suggested approach seems to be a step back as being less accurate and significantly more lengthy.

Minor comment:

1. Why is Ernie (one model) compared to the entire HCP dataset?

7. PLOS authors have the option to publish the peer review history of their article (what does this mean?). If published, this will include your full peer review and any attached files.

Reviewer #2: No

Reviewer #3: No

Reviewer #4: No

---

## [Author Response · Author response to Decision Letter 1]

23 Jun 2021

See the attached response-to-reviewers letter.

---

## [Editor Report · Decision Letter 2]

30 Jun 2021

Rapid whole-brain electric field mapping in transcranial magnetic stimulation using deep learning

PONE-D-20-30151R2

Dear Dr. Ning,

We’re pleased to inform you that your manuscript has been judged scientifically suitable for publication and will be formally accepted for publication once it meets all outstanding technical requirements.

Kind regards,

Dzung Pham

Academic Editor

PLOS ONE
---

## [Editor Report · Acceptance letter]

21 Jul 2021

PONE-D-20-30151R2 

Rapid whole-brain electric field mapping in transcranial magnetic stimulation using deep learning 

Dear Dr. Ning:

I'm pleased to inform you that your manuscript has been deemed suitable for publication in PLOS ONE. Congratulations! Your manuscript is now with our production department. 

Kind regards, 

on behalf of

Dr Dzung Pham 

Academic Editor

PLOS ONE